# Efficient Learning by Directed Acyclic Graph For Resource Constrained Prediction

**Joseph Wang**
Department of Electrical
& Computer Engineering
Boston University,
Boston, MA 02215
joewang@bu.edu

**Kirill Trapeznikov**
Systems & Technology Research
Woburn, MA 01801
kirill.trapeznikov@
stresearch.com

**Venkatesh Saligrama**
Department of Electrical
& Computer Engineering
Boston University,
Boston, MA 02215
srv@bu.edu

## Abstract

We study the problem of reducing test-time acquisition costs in classification systems. Our goal is to learn decision rules that adaptively select sensors for each example as necessary to make a confident prediction. We model our system as a directed acyclic graph (DAG) where internal nodes correspond to sensor subsets and decision functions at each node choose whether to acquire a new sensor or classify using the available measurements. This problem can be posed as an empirical risk minimization over training data. Rather than jointly optimizing such a highly coupled and non-convex problem over all decision nodes, we propose an efficient algorithm motivated by dynamic programming. We learn node policies in the DAG by reducing the global objective to a series of cost sensitive learning problems. Our approach is computationally efficient and has proven guarantees of convergence to the optimal system for a fixed architecture. In addition, we present an extension to map other budgeted learning problems with large number of sensors to our DAG architecture and demonstrate empirical performance exceeding state-of-the-art algorithms for data composed of both few and many sensors.

## 1 Introduction

Many scenarios involve classification systems constrained by measurement acquisition budget. In this setting, a collection of sensor modalities with varying costs are available to the decision system. Our goal is to learn adaptive decision rules from labeled training data that, when presented with an unseen example, would select the most informative and cost-effective acquisition strategy for this example. In contrast, non-adaptive methods [24] attempt to identify a common sparse subset of sensors that can work well for all data. Our goal is an adaptive method that can classify typical cases using inexpensive sensors while using expensive sensors only for atypical cases.

We propose an adaptive sensor acquisition system learned using labeled training examples. The system, modeled as a directed acyclic graph (DAG), is composed of internal nodes, which contain decision functions, and a single sink node (the only node with no outgoing edges), representing the terminal action of stopping and classifying (SC). At each internal node, a decision function routes an example along one of the outgoing edges. Sending an example to another internal node represents acquisition of a previously unacquired sensor, whereas sending an example to the sink node indicates that the example should be classified using the currently acquired set of sensors. The goal is to learn these decision functions such that the expected error of the system is minimized subject to an expected budget constraint.

First, we consider the case where the number of sensors available is small (as in [19, 23, 20]), though the dimensionality of data acquired by each sensor may be large (such as an image taken in different

modalities). In this scenario, we construct a DAG that allows for sensors to be acquired in any order and classification to occur with any set of sensors. In this regime, we propose a novel algorithm to learn node decisions in the DAG by emulating dynamic programming (DP). In our approach, we decouple a complex sequential decision problem into a series of tractable cost-sensitive learning subproblems. Cost-sensitive learning (CSL) generalizes multi-decision learning by allowing decision costs to be data dependent [2]. Such reduction enables us to employ computationally efficient CSL algorithms for iteratively learning node functions in the DAG. In our theoretical analysis, we show that, given a fixed DAG architecture, the policy risk learned by our algorithm converges to the Bayes risk as the size of the training set grows.

Next, we extend our formulation to the case where a large number of sensors exist, but the number of distinct sensor subsets that are necessary for classification is small (as in [25, 11] where the depth of the trees is fixed to 5). For this regime, we present an efficient subset selection algorithm based on sub-modular approximation. We treat each sensor subset as a new "sensor," construct a DAG over unions of these subsets, and apply our DP algorithm. Empirically, we show that our approach outperforms state-of-the-art methods in both small and large scale settings.

**Related Work:** There is an extensive literature on adaptive methods for sensor selection for reducing test-time costs. It arguably originated with detection cascades (see [26, 4] and references therein), a popular method in reducing computation cost in object detection for cases with highly skewed class imbalance and generic features. Computationally cheap features are used at first to filter out negative examples and more expensive features are used in later stages.

Our technical approach is closely related to Trapeznikov et al. [19] and Wang et al. [23, 20]. Like us they formulate an ERM problem and generalize detection cascades to classifier cascades and trees and handle balanced and/or multi-class scenarios. Trapeznikov et al. [19] propose a similar training scheme for the case of cascades, however restrict their training to cascades and simple decision functions which require alternating optimization to learn. Alternatively, Wang et al. [21, 22, 23, 20] attempt to jointly solve the decision learning problem by formulating a linear upper-bounding surrogate, converting the problem into a linear program (LP).

Conceptually, our work is closely related to Xu et al. [25] and Kusner et al.[11], who introduce Cost-Sensitive Trees of Classifiers (CSTC) and Approximately Submodular Trees of Classifiers (ASTC), respectively, to reducing test time costs. Like our paper they propose a global ERM problem. They solve for the tree structure, internal decision rules and leaf classifiers jointly using alternative minimization techniques. Recently, Kusner et al.[11] propose Approximately Submodular Trees of Classifiers (ASTC), a variation of CSTC which provides robust performance with significantly reduced training time and greedy approximation, respectively. Recently, Nan et al. [14] proposed random forests to efficiently learn budgeted systems using greedy approximation over large data sets.

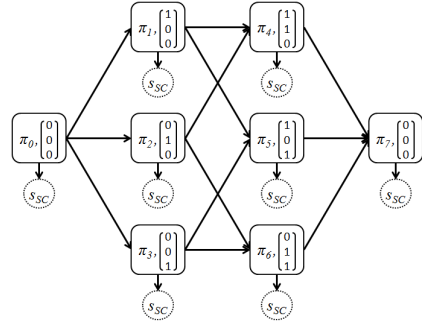

Figure 1: A simple example of a sensor selection DAG for a three sensor system. At each state, represented by a binary vector indicating measured sensors, a policy $\pi$ chooses between either adding a new sensor or stopping and classifying. Note that the state $s_{SC}$ has been repeated for simplicity.

The subject of this paper is broadly related to other adaptive methods in the literature. Generative methods [17, 8, 9, 6] pose the problem as a POMDP, learn conditional probability models, and myopically select features based information gain of unknown features. MDP-based methods [5, 10, 7, 3] encode current observations as state, unused features as action space, and formulate various reward functions to account for classification error and costs. He et al. [7] apply imitation learning of a greedy policy with a single classification step as actions. Dulac-Arnold et al. [5] and Karayev et al. [10] apply reinforcement learning to solve this MDP. Benbouzid et al.[3] propose classifier cascades with an additional skip action within an MDP framework. Nan et al. [15] consider a nearest neighbor approach to feature selection, with confidence driven by margin magnitude.

## 2 Adaptive Sensor Acquisition by DAG

In this section, we present our adaptive sensor acquisition DAG that during test-time sequentially decides which sensors should be acquired for every new example entering the system. Before formally describing the system and our learning approach, we first provide a simple illustration for a 3 sensor DAG shown in Fig. 1. The state indicating acquired sensors is represented by a binary vector, with a 0 indicating that a sensor measurement has not been acquired and a 1 representing an acquisition. Consider a new example that enters the system. Initially, it has a state of $[0, 0, 0]^T$ (as do all samples during test-time) since no sensors have been acquired. It is routed to the policy function $\pi_0$, which makes a decision to measure one of the three sensors or to stop and classify. Let us assume that the function $\pi_0$ routes the example to the state $[1, 0, 0]^T$, indicating that the first sensor is acquired. At this node, the function $\pi_1$ has to decide whether to acquire the second sensor, acquire the third, or classifying using only the first. If $\pi_1$ chooses to stop and classify then this example will be classified using only the first sensor.

Such decision process is performed for every new example. The system adaptively collects sensors until the policy chooses to stop and classify (we assume that when all sensors have been collected the decision function has no choice but to stop and classify, as shown for $\pi_7$ in Fig. 1).

**Problem Formulation:** A data instance, $x \in \mathcal{X}$, consists of $M$ sensor measurements, $x = \{x^1, x^2, \ldots, x^M\}$, and belongs to one of $L$ classes indicated by its label $y \in \mathcal{Y} = \{1, 2, \ldots L\}$. Each sensor measurement, $x^m$, is not necessarily a scalar but may instead be multi-dimensional. Let the pair, $(x, y)$, be distributed according to an unknown joint distribution $\mathcal{D}$. Additionally, associated with each sensor measurement $x^m$ is an acquisition cost, $c_m$.

To model the acquisition process, we define a state space $\mathcal{S} = \{s_1, \ldots, s_K, s_{SC}\}$. The states $\{s_1, \ldots, s_K\}$ represent subsets of sensors, and the stop-and-classify state $s_{SC}$ represents the action of stopping and classifying with a current subset. Let $\mathcal{X}_s$ correspond to the space of sensor measurements in subset $s$. We assume that the state space includes all possible subsets[1], $K = 2^M$. For example in Fig. 1, the system contains all subsets of 3 sensors. We also introduce the state transition function, $\mathcal{T} : \mathcal{S} \to \mathcal{S}$, that defines a set of actions that can be taken from the current state. A transition from the current sensor subset to a new subset corresponds to an acquisition of new sensor measurements. A transition to the state $s_{SC}$ corresponds to stopping and classifying using the available information. This terminal state, $s_{SC}$, has access to a classifier bank used to predict the label of an example. Since classification has to operate on any sensor subset, there is one classifier for every $s_k$: $f_{s_1}, \ldots, f_{s_K}$ such that $f_s : \mathcal{X}_s \to \mathcal{Y}$. We assume the classifier bank is given and pre-trained. Practically, the classifiers can be either unique for each subset or a missing feature (i.e. sensor) classification system as in [13]. We overload notation and use node, subset of sensors, and path leading up to that subset on the DAG interchangeably. In particular we let $\mathcal{S}$ denote the collection of subsets of nodes. Each subset is associated with a node on the DAG graph. We refer to each node as a state since it represents the "state-of-information" for an instance at that node.

We define the loss associated with classifying an example/label pair $(x, y)$ using the sensors in $s_j$ as

$$L_{s_j}(x, y) = \mathbb{1}_{f_{s_j}(x) \neq y} + \sum_{k \in s_j} c_k. \tag{1}$$

Using this convention, the loss is the sum of the empirical risk associated with classifier $f_{s_j}$ and the cost of the sensors in the subset $s_j$. The expected loss over the data is defined

$$\mathcal{L}_{\mathcal{D}}(\pi) = E_{x, y \sim \mathcal{D}} \left[ L_{\pi(x)}(x, y) \right]. \tag{2}$$

Our goal is to find a policy which adaptively selects subsets for examples such that their average loss is minimized

$$\min_{\pi \in \Pi} \mathcal{L}_{\mathcal{D}}(\pi), \tag{3}$$

where $\pi : \mathcal{X} \to \mathcal{S}$ is a policy selected from a family of policies $\Pi$ and $\pi(x)$ is the state selected by the policy $\pi$ for example $x$. We denote the quantity $\mathcal{L}_{\mathcal{D}}$ as the value of (3) when $\Pi$ is the family of all measurable functions. $\mathcal{L}_{\mathcal{D}}$ is the *Bayes cost*, representing the minimum possible cost for any

function given the distribution of data. In practice, the distribution $\mathcal{D}$ is unknown, and instead we are given training examples $(x_1, y_1), \ldots, (x_n, y_n)$ drawn I.I.D. from $\mathcal{D}$. The problem becomes an empirical risk minimization:

$$\min_{\pi \in \Pi} \sum_{i=1}^{n} L_{\pi(x_i)}(x_i, y_i). \tag{4}$$

Recall that our sensor acquisition system is represented as a DAG. Each node in a graph corresponds to a state (i.e. sensor subset) in $\mathcal{S}$, and the state transition function, $\mathcal{T}(s_j)$, defines the outgoing edges from every node $s_j$. We refer to the entire edge set in the DAG as $E$. In such a system, the policy $\pi$ is parameterized by the set of decision functions $\pi_1, \ldots, \pi_K$ at every node in the DAG. Each function, $\pi_j : \mathcal{X} \to \mathcal{T}(s_j)$, maps an example to a new state (node) from the set specified by outgoing edges. Rather than directly minimizing the empirical risk in (4), first, we define a step-wise cost associated with all edges $(s_j, s_k) \in E$

$$C(x, y, s_j, s_k) = \begin{cases} \sum_{t \in s_k \setminus s_j} c_t & \text{if } s_k \neq s_{SC} \\ \mathbb{1}_{f_{s_j}(x) \neq y} & \text{otherwise} \end{cases}. \tag{5}$$

$C(\cdot)$ is either the cost of acquiring new sensors or is the classification error induced by classifying with the current subset if $s_k = s_{SC}$. Using this step-wise cost, we define the empirical loss of the system w.r.t a path for an example $x$:

$$R(x, y, \pi_1, ..., \pi_K) = \sum_{(s_j, s_{j+1}) \in \text{path}(x, \pi_1, ..., \pi_K)} C(x, y, s_j, s_{j+1}), \tag{6}$$

where path $(x, \pi_1, \ldots, \pi_K)$ is the path on the DAG induced by the policy functions $\pi_1, \ldots, \pi_K$ for example $x$. The empirical minimization equivalent to (4) for our DAG system is a sample average over all example specific path losses:

$$\pi_1^*, \ldots, \pi_K^* = \operatorname*{argmin}_{\pi_1, \ldots, \pi_K \in \Pi} \sum_{i=1}^{n} R(x_i, y_i, \pi_1, \ldots, \pi_K). \tag{7}$$

Next, we present a reduction to learn the functions $\pi_1, ..., \pi_K$ that minimize the loss in (7).

## 2.1 Learning Policies in a DAG

Learning the functions $\pi_1, \ldots, \pi_K$ that minimize the cost in (7) is a highly coupled problem. Learning a decision function $\pi_j$ is dependent on the other functions in two ways: (**a**) $\pi_j$ is dependent on functions at nodes downstream (nodes for which a path exists from $\pi_j$), as these determine the cost of each action taken by $\pi_j$ on an individual example (the cost-to-go), and (**b**) $\pi_j$ is dependent on functions at nodes upstream (nodes for which a path exists to $\pi_j$), as these determine the distribution of examples that $\pi_j$ acts on. Consider a policy $\pi_j$ at a node corresponding to state $s_j$ such that all outgoing edges from $j$ lead to leaves. Also, we assume all examples pass through this node $\pi_j$ (we are ignoring the effect of upstream dependence **b**). This yields the following important lemma:

**Lemma 2.1.** *Given the assumptions above, the problem of minimizing the risk in* (6) *w.r.t a single policy function, $\pi_j$, is equivalent to solving a k-class cost sensitive learning (CSL) problem.*[2]

*Proof.* Consider the risk in (6) with $\pi_j$ such that all outgoing edges from $j$ lead to a leaf. Ignoring the effect of other policy functions upstream from $j$, the risk w.r.t $\pi_j$ is:

$$R(x, y, \pi_j) = \sum_{s_k \in \mathcal{T}(s_j)} C(x, y, s_j, s_k) \mathbb{1}_{\pi_j(x) = s_k} \to \min_{\pi \in \Pi} \sum_{i=1}^{n} R(x_i, y_i, \pi_j).$$

Minimizing the risk over training examples yields the optimization problem on the right hand side. This is equivalent to a CSL problem over the space of "labels" $\mathcal{T}(s_j)$ with costs given by the transition costs $C(x, y, s_j, s_k)$. □

In order to learn the policy functions $\pi_1, \ldots, \pi_K$, we propose Algorithm 1, which iteratively learns policy functions using Lemma 2.1. We solve the CSL problem by using a filter-tree scheme [2] for $Learn$, which constructs a tree of binary classifiers. Each binary classifier can be trained using regularized risk minimization. For concreteness we define the $Learn$ algorithm as

$$Learn((x_1, \vec{w}_1), ..., (x_n, \vec{w}_n)) \triangleq$$
$$FilterTree((x_1, \vec{w}_1), ..., (x_n, \vec{w}_n)) \tag{8}$$

where the binary classifiers in the filter tree are trained using an appropriately regularized calibrated convex loss function. Note that multiple schemes exist that map the CSL problem to binary classification.

A single iteration of Algorithm 1 proceeds as follows: (**1**) A node $j$ is chosen whose outgoing edges connect only to leaf nodes. (**2**) The costs associated with each connected leaf node are found. (**3**) The policy $\pi_j$ is trained on the entire set of training data according to these costs by solving a CSL problem. (**4**) The costs associated with taking the action $\pi_j$ are computed for each example, and the costs of moving to state $j$ are updated. (**5**) Outgoing edges from node $j$ are removed (making it a leaf node), and (**6**) disconnected nodes (that were previously connected to node $j$) are removed. The algorithm iterates through these steps until all edges have been removed. We denote the policy functions trained on the empirical data using Alg. 1 as $\pi_1^n, \ldots, \pi_K^n$.

---

**Algorithm 1** Graph Reduce Algorithm

---

**Input:** Data: $(x_i, y_i)_{i=1}^n$,
DAG: (nodes $\mathcal{S}$, edges $E$, costs $C(x_i, y_i, e), \forall e \in E$),
CSL alg: $Learn\left((x_1, \vec{w}_1), \ldots, (x_n, \vec{w}_n)\right)) \rightarrow \pi(\cdot)$
**while** Graph $\mathcal{S}$ is NOT empty **do**
    (**1**) Choose a node, $j \in \mathcal{S}$, s.t. all children of $j$ are leaf nodes
    **for** example $i \in \{1, \ldots, n\}$ **do**
        (**2**) Construct the weight vector $\vec{w}_i$ of edge costs per action.
    **end for**
    (**3**) $\pi_j \leftarrow Learn\left((x_1, \vec{w}_1), \ldots, (x_n, \vec{w}_n)\right)$
    (**4**) Evaluate $\pi_j$ and update edge costs to node $j$:
    $C(x_i, y_i, s_n, s_j) \leftarrow \vec{w}_i^j\left(\pi_j(x_i)\right) + C(x_i, y_i, s_n, s_j)$
    (**5**) Remove all outgoing edges from node $j$ in $E$
    (**6**) Remove all disconnected nodes from $\mathcal{S}$.
**end while**
**Output:** Policy functions, $\pi_1, \ldots, \pi_K$

---

## 2.2 Analysis

Our goal is to show that the expected risk of the policy functions $\pi_1, \ldots, \pi_K$ learned by Alg. 1 converge to the Bayes risk. We first state our main result:

**Theorem 2.2.** *Alg. 1 is universally consistent, that is*

$$\lim_{n \rightarrow \infty} \mathcal{L}_\mathcal{D}(\pi_1^n, \ldots, \pi_K^n) \rightarrow \mathcal{L}_\mathcal{D} \qquad (9)$$

*where $\pi_1^n, \ldots, \pi_K^n$ are the policy functions learned using Alg.* (1)*, which in turn uses Learn described by Eq. 8.*

Alg. 1 emulates a dynamic program applied in an empirical setting. Policy functions are decoupled and trained from leaf to root conditioned on the output of descendant nodes.

To adapt to the empirical setting, we optimize at each stage over all examples in the training set. The key insight is the fact that universally consistent learners output optimal decisions over subsets of the space of data, that is they are locally optimal. To illustrate this point, consider a standard classification problem. Let $\mathcal{X}' \subset \mathcal{X}$ be the support (or region) of examples induced by upstream deterministic decisions. $d^*$ and $f^*$, Bayes optimal classifiers w.r.t the full space and subset, respectively, are equal on the reduced support:

$$d^*(x) = \arg\min_d E\left[\mathbb{1}_{d(x) \neq y} | x\right] = f^*(x) = \arg\min_f E\left[\mathbb{1}_{f(x) \neq y} | x, x \in \mathcal{X}' \subset \mathcal{X}\right] \ \forall \ x \in \mathcal{X}'.$$

From this insight, we decouple learning problems while still training a system that converges to the Bayes risk. This can be achieved by training universally consistent CSL algorithms such as filter trees [2] that reduce the problem to binary classification. By learning consistent binary classifiers [1, 18], the risk of the cost-sensitive function can be shown to converge to the Bayes risk [2]. Proof of Theorem 2.2 is included in the Supplementary Material.

**Computational Efficiency:** Alg. 1 reduces the problem to solving a series of $O(KM)$ binary classification problems, where $K$ is the number of nodes in the DAG and $M$ is the number of sensors. Finding each binary classifier is computationally efficient, reducing to a convex problem with $O(n)$ variables. In contrast, nearly all previous approaches require solving a non-convex problem and resort to alternating optimization [25, 19] or greedy approximation [11]. Alternatively, convex surrogates proposed for the global problem [23, 20] require solving large convex programs with $\theta(n)$ variables, even for simple linear decision functions. Furthermore, existing off-the-shelf algorithms cannot be applied to train these systems, often leading to less efficient implementation.

## 2.3 Generalization to Other Budgeted Learning Problems

Although, we presented our algorithm in the context of supervised classification and a uniform linear sensor acquisition cost structure, the above framework holds for a wide range of problems.

In particular, any loss-based learning problem can be solved using the proposed DAG approach by generalizing the cost function

$$\tilde{C}(x, y, s_j, s_k) = \begin{cases} c(x, y, s_j, s_k) & \text{if } s_k \neq s_{SC} \\ D(x, y, s_j) & \text{otherwise} \end{cases}, \tag{10}$$

where $c(x, y, s_j, s_k)$ is the cost of acquiring sensors in $s_k \backslash s_j$ for example $(x, y)$ given the current state $s_j$ and $D(x, y, s_j)$ is some loss associated with applying sensor subset $s_j$ to example $(x, y)$. This framework allows for significantly more complex budgeted learning problems to be handled. For example, the sensor acquisition cost, $c(x, y, s_j, s_k)$, can be object dependent and non-linear, such as increasing acquisition costs as time increases (which can arise in image retrieval problems, where users are less likely to wait as time increases). The cost $D(x, y, s_j)$ can include alternative costs such as $\ell_2$ error in regression, precision error in ranking, or model error in structured learning. As in the supervised learning case, the learning functions and example labels do not need to be explicitly known. Instead, the system requires only empirical performance to be provided, allowing complex decision systems (such as humans) to be characterized or systems learned where the classifiers and labels are sensitive information.

## 3 Adaptive Sensor Acquisition in High-Dimensions

So far, we considered the case where the DAG system allows for any subset of sensors to be acquired, however this is often computationally intractable as the number of nodes in the graph grows exponentially with the number of sensors. In practice, these complete systems are only feasible for data generated from a small set of sensors ( 10 or less).

### 3.1 Learning Sensor Subsets

Although constructing an exhaustive DAG for data with a large number of sensors is computationally intractable, in many cases this is unnecessary. Motivated by previous methods [6, 25, 11], we assume that the number of "active" nodes in the exhaustive graph is small, that is these nodes are either not visited by any examples or all examples that visit the node acquire the same next sensor. Equivalently, this can be viewed as the system needing only a small number of sensor subsets to classify all examples with low acquisition cost.

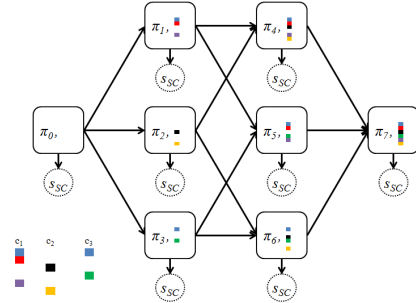

Figure 2: An example of a DAG system using the 3 sensor subsets shown on the bottom left. The new states are the union of these sensor subsets, with the system otherwise constructed in the same fashion as the small scale system.

Rather than attempt to build the entire combinatorially sized graph, we instead use this assumption to first find these "active" subsets of sensors and construct a DAG to choose between unions of these subsets. The step of finding these sensor subsets can be viewed as a form of feature clustering, with a goal of grouping features that are jointly useful for classification. By doing so, the size of the DAG is reduced from exponential in the number of sensors, $2^M$, to exponential in a much smaller user chosen parameter number of subsets, $2^t$. In experimental results, we limit $t = 8$, which allows for a diverse subsets of sensors to be found while preserving computational tractability and efficiency.

Our goal is to learn sensor subsets with high classification performance and low acquisition cost (empirically low cost as defined in (1)). Ideally, our goal is to jointly learn the subsets which minimize the empirical risk of the entire system as defined in (4), however this presents a computationally intractable problem due to the exponential search space. Rather than attempt to solve this difficult problem directly, we minimize classification error over a collection of sensor subsets $\sigma_1, \ldots, \sigma_t$ subject to a cost constraint on the total number of sensors used. We decouple the problem from the policy learning problem by assuming that each example is classified by the best possible subset. For a constant sensor cost, the problem can be expressed as a set constraint problem:

$$\min_{\sigma_1, \ldots, \sigma_t} \frac{1}{N} \sum_{i=1}^{N} \min_{j \in \{1, \ldots, t\}} \left[ \mathbb{1}_{f_{\sigma_j}(x_i) \neq y_i} \right] \quad \text{such that: } \sum_{j=1}^{t} |\sigma_j| \leq \frac{B}{\delta}, \tag{11}$$

where $B$ is the total sensor budget over all sensor subsets and $\delta$ is the cost of a single sensor.

Although minimizing this loss is still computationally intractable, consider instead the equivalent problem of maximizing the "reward" (the event of a correct classification) of the subsets, defined as

$$G = \sum_{i=1}^{N} \max_{j \in \{1,\ldots,t\}} \left[ \mathbb{1}_{f_{\sigma_j}(x_i)=y_i} \right] \rightarrow \max_{\sigma_1,\ldots,\sigma_t} \frac{1}{N} G(c_1,\ldots,c_t) \text{ such that: } \sum_{j=1}^{t} |\sigma_j| \leq \frac{B}{\delta}. \qquad (12)$$

This problem is related to the knapsack problem with a non-linear objective. Maximizing the reward in (12) is still a computationally intractable problem, however the reward function is structured to allow for efficient approximation.

**Lemma 3.1.** *The objective of the maximization in* (12) *is sub-modular with respect to the set of subsets, such that adding any new set to the reward yields diminishing returns.*

**Theorem 3.2.** *Given that the empirical risk of each classifier $f_{\sigma_k}$ is submodular and monotonically decreasing w.r.t. the elements in $\sigma_k$ and uniform sensor costs, the strategy in Alg. 2 is an $O(1)$ approximation of the optimal reward in* (12).

Proof of these statements is included in the Supplementary Material and centers on showing that the objective is sub-modular, and therefore applying a greedy strategy yields a $1 - \frac{1}{e}$ approximation of the optimal strategy [16].

### 3.2 Constructing DAG using Sensor Subsets

Alg. 2 requires computation of the reward $G$ for only $O\left(\frac{B}{\delta} t M\right)$ sensor subsets, where $M$ is the number of sensors, to return a constant-order approximation to the NP-hard knapsack-type problem. Given the set of sensor subsets $\sigma_1, \ldots, \sigma_t$, we can now construct a DAG using all possible unions of these subsets, where each sensor subset $\sigma_j$ is treated as a new single sensor, and apply the small scale system presented in Sec. 2. The result is an efficiently learned system with relatively low complexity yet strong performance/cost trade-off. Additionally, this result can be extended to the case of non-uniform costs, where a simple extension of the greedy algorithm yields a constant-order approximation [12].

---

**Algorithm 2** Sensor Subset Selection

**Input:** Number of Subsets $t$, Cost Constraint $\frac{B}{\delta}$

**Output:** Feature subsets, $\sigma_1, \ldots, \sigma_t$

**Initialize:** $\sigma_1, \ldots, \sigma_t = \emptyset$

$(i,j) = \text{argmax}_{i \in \{1,\ldots,t\}} \text{argmax}_{j \in \sigma_i^C} G(\sigma_1, ..., \sigma_i \cup j, ..., \sigma_t)$

**while** $\sum_{j=1}^{T} |\sigma_j| \leq \frac{C}{\delta}$ **do**

$\quad \sigma_i = \sigma_i \cup j$

$\quad (i,j) = \text{argmax}_{i \in \{1,\ldots,t\}} \text{argmax}_{j \in \sigma_i^C} G(\sigma_1, ..., \sigma_i \cup j, ..., \sigma_t)$

**end while**

---

A simple case where three subsets are used is shown in Fig. 2. The three learned subsets of sensors are shown on the bottom left of Fig. 2, and these three subsets are then used to construct the entire DAG in the same fashion as in Fig. 1. At each stage, the state is represented by the union of sensor subsets acquired. Grouping the sensors in this fashion reduces the size of the graph to 8 nodes as opposed to 64 nodes required if any subset of the 6 sensors can be selected. This approach allows us to map high-dimensional adaptive sensor selection problems to small scale DAG in Sec. 2.

## 4 Experimental Results

To demonstrate the performance of our DAG sensor acquisition system, we provide experimental results on data sets previously used in budgeted learning. Three data sets previously used for budget cascades [19, 23] are tested. In these data sets, examples are composed of a small number of sensors (under 4 sensors). To compare performance, we apply the LP approach to learning sensor trees [20] and construct trees containing all subsets of sensors as opposed to fixed order cascades [19, 23].

Next, we examine performance of the DAG system using 3 higher dimensional sets of data previously used to compare budgeted learning performance [11]. In these cases, the dimensionality of the data (between 50 and 400 features) makes exhaustive subset construction computationally infeasible. We greedily construct sensor subsets using Alg. 2, then learn a DAG over all unions of these sensor subsets. We compare performance with CSTC [25] and ASTC [11].

For all experiments, we use cost sensitive filter trees [2], where each binary classifier in the tree is learned using logistic regression. Homogeneous polynomials are used as decision functions in the filter trees. For all experiments, uniform sensor cost were were varied in the range $[0, M]$ achieve systems with different budgets. Performance between the systems is compared by plotting the average number of features acquired during test-time vs. the average test error.

## 4.1 Small Sensor Set Experiments

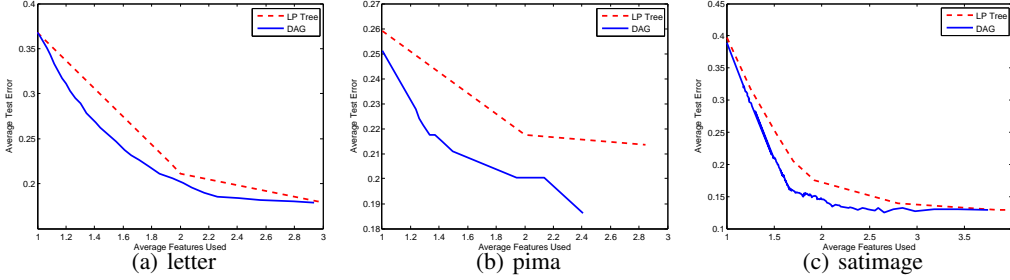

Figure 3: Average number of sensors acquired vs. average test error comparison between LP tree systems and DAG systems.

We compare performance of our trained DAG with that of a complete tree trained using an LP surrogate [20] on the landsat, pima, and letter datasets. To construct each sensor DAG, we include all subsets of sensors (including the empty set) and connect any two nodes differing by a single sensor, with the edge directed from the smaller sensor subset to the larger sensor subset. By including the empty set, no initial sensor needs to be selected. $3^{rd}$-order homogeneous polynomials are used for both the classification and system functions in the LP and DAG.

As seen in Fig. 3, the systems learned with a DAG outperform the LP tree systems. Additionally, the performance of both of the systems is significantly better than previously reported performance on these data sets for budget cascades [19, 23]. This arises due to both the higher complexity of the classifiers and decision functions as well as the flexibility of sensor acquisition order in the DAG and LP tree compared to cascade structures. For this setting, it appears that the DAG approach is superior approach to LP trees for learning budgeted systems.

## 4.2 Large Sensor Set Experiments

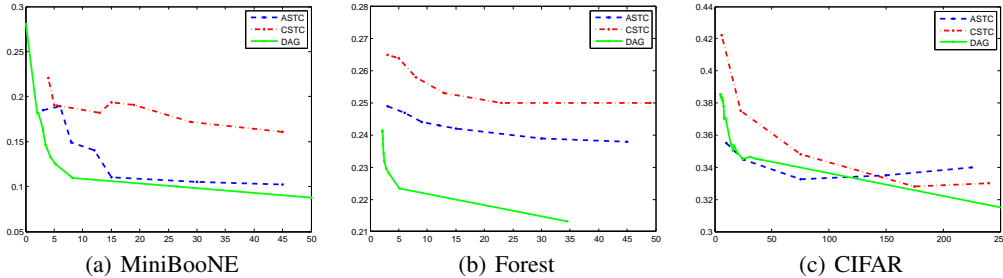

Figure 4: Comparison between CSTC, ASTC, and DAG of the average number of acquired features (x-axis) vs. test error (y-axis).

Next, we compare performance of our trained DAG with that of CSTC [25] and ASTC [11] for the MiniBooNE, Forest, and CIFAR datasets. We use the validation data to find the homogeneous polynomial that gives the best classification performance using all features (MiniBooNE: linear, Forest: $2^{nd}$ order, CIFAR: $3^{rd}$ order). These polynomial functions are then used for all classification and policy functions. For each data set, Alg. 2 was used to find 7 subsets, with an $8^{th}$ subset of all features added. An exhaustive DAG was trained over all unions of these 8 subsets.

Fig. 4 shows performance comparing the average cost vs. average error of CSTC, ASTC, and our DAG system. The systems learned with a DAG outperform both CSTC and ASTC on the Mini-BooNE and Forest data sets, with comparable performance on CIFAR at low budgets and superior performance at higher budgets.

### Acknowledgments

This material is based upon work supported in part by the U.S. National Science Foundation Grant 1330008, by the Department of Homeland Security, Science and Technology Directorate, Office of University Programs, under Grant Award 2013- ST-061-ED0001, by ONR Grant 50202168 and US AF contract FA8650-14-C-1728. The views and conclusions contained in this document are those of the authors and should not be interpreted as necessarily representing the social policies, either expressed or implied, of the U.S. DHS, ONR or AF.

## Footnotes

[1]While enumerating all possible combinations is feasible for small $M$, for large $M$ this problem becomes intractable. We will overcome this limitation in Section 3 by applying a novel sensor selection algorithm. For now, we remain in the small $M$ regime.

[2]We consider the k-class CSL problem formulated by Beygelzimer et al. [2], where an instance of the problem is defined by a distribution $D$ over $\mathcal{X} \times [0, \inf)^k$, a space of features and associated costs for predicting each of the $k$ labels for each realization of features. The goal is to learn a function which maps each element of $\mathcal{X}$ to a label $\{1, \ldots, k\}$ s.t. the expected cost is minimized.

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
