[Supplementary Material]

# Efficient Learning by Directed Acyclic Graph For Resource Constrained Prediction: Supplementary Material

**Joseph Wang**
Department of Electrical
& Computer Engineering
Boston University,
Boston, MA 02215
joewang@bu.edu

**Kirill Trapeznikov**
Systems & Technology Research
Woburn, MA 01801
kirill.trapeznikov@stresearch.com

**Venkatesh Saligrama**
Department of Electrical
& Computer Engineering
Boston University,
Boston, MA 02215
srv@bu.edu

## 1 Proof of Theorem 2.2

**Theorem 2.2** Alg. 1 is universally consistent, that is

$$\lim_{n \to \infty} \mathcal{L}_{\mathcal{D}}(\pi_1^n, \ldots, \pi_K^n) \to \mathcal{L}_{\mathcal{D}} \tag{1}$$

where $\pi_1^n, \ldots, \pi_K^n$ are the policy functions learned using Alg. 1, which in turn uses $Learn$ described by Eq. 8.

*Proof.* The proof can be broken down into two steps. First, we show that training the policy function with no downstream policy functions decouples from other policies. Next, we show that sequentially learning policy functions from leaf to root leads to an optimal policy.

Consider first the node associated with state $s_j$ whose outgoing edges lead to leaves. Alg. 1 trains the policy $\pi_j^n$ over the entire training set using (8). As (8) is a universally consistent algorithm, the $\pi_j^n$ converges to the optimal policy as the data grows:

$$\lim_{n \to \infty} \mathbb{E}_{x,y,\sim \mathcal{D}} \left[ C(x, y, s_j, \pi_j^n(x)) \right] \to \inf \{ \mathbb{E}_{x,y,\sim \mathcal{D}} \left[ C(x, y, s_j, \pi_j^*(x)) \right] | \pi_j^* : \mathcal{X} \to \mathcal{S} \}$$

where the infimum is over any measurable function $\pi_j^*$. As this infimum is over any measurable function, we point out that this convergence holds for any realization $x \in \mathcal{X}^j$

$$\mathbb{E}_{y \sim \mathcal{D}(x)} \left[ C(x, y, s_j, \pi_j^n(x)) | x \right] \to \inf \{ \mathbb{E}_{y \sim \mathcal{D}(x)} \left[ C(x, y, s_j, \pi_j^*(x)) | x \right] | \pi_j^* : \mathcal{X} \to \mathcal{S} \}$$

where $\mathcal{D}(x)$ is the distribution of $y$ conditioned on $x$. As the outgoing edges of node $s_j$ contain only leaves, other policy functions $\pi_1^n, \ldots, \pi_{j-1}^n, \pi_{j+1}^n, \ldots, \pi_K^n$ do not affect the conditional distribution $\mathcal{D}(x)$. Instead, they only reduce the support of $\mathcal{X}$ observed by $\pi_j^n$, and therefore $\pi_j^n$ converges to the Bayes optimal function independent of the other policy functions.

Alg. 1 updates the edge costs (costs-to-go) of edges directed to $s_j$. These costs-to-go do not vary as we train new policy functions in ancestor nodes of $s_j$ and these values can be viewed as fixed when training the next policy function, $\pi_k^n$. The dependence on $\pi_j^n$ is well captured when learning $\pi_k^n$. As $\pi_j^n$ converges to the Bayes optimal function, the costs-to-go converge to the Bayes optimal values, $\pi_k^n$ is trained on the Bayes optimal costs-to-go as $n \to \infty$.

By recursion, this implies that every decision function is learned on costs-to-go approaching the Bayes optimal, and therefore each function approaches the point-wise Bayesian optimal decision. Consequently, the learned system approaches the Bayesian optimal system. □

## 2 Proof of Lemma 3.1 and Theorem 3.2

For convenience, we repeat the subset optimization problem.

$$G = \sum_{i=1}^{N} \max_{j \in \{1, \ldots, t\}} \left[ \mathbb{1}_{f_{\sigma_j}(x_i) = y_i} \right] \rightarrow \max_{\sigma_1, \ldots, \sigma_t} \frac{1}{N} G(c_1, \ldots, c_t) \text{ such that: } \sum_{j=1}^{t} |\sigma_j| \leq \frac{B}{\delta}. \tag{2}$$

**Lemma 3.1** *The objective of the maximization in* (2) *is submodular with respect to the set of subsets, such that adding any new set to the reward yields diminishing returns.*

*Proof.* This follows directly from the fact that maximization over a set of objects is a submodular function. □

**Theorem 3.2** *Given that the empirical risk of each classifier $f_{\sigma_k}$ is submodular and monotonically decreasing w.r.t the elements in $\sigma_k$ and uniform costs among sensors, the strategy in Alg. 2 is an $O(1)$ approximation of the optimal reward in* (2).

*Proof.* Consider adding a sensor $k$ to any subset $\sigma_j$. By assumption, the empirical risk of each classifier is monotonically decreasing and therefore the reward is monotonically increasing. Additionally, note that the reward for any training point $x_i$ using $\sigma_j$ is less than the reward from using $\sigma_j \cup k$ and therefore the objective is equal to the objective without replacement of $\sigma_j$ by $\sigma_j \cup k$:

$$G(c_1, \ldots, c_{j-1}, c_j \cup k, \ldots, c_K) = G(c_1, \ldots, c_j, c_j \cup k, \ldots, c_K).$$

As a result, we can view adding a sensor to a subset as adding an entirely new subset without changing the objective in (2). From the above lemma, adding a new subset results in a submodular function, and therefore the reward in (2) is submodular with respect to adding sensors to each subset. Applying a greedy strategy therefore yields a $1 - \frac{1}{e}$ approximation of the optimal strategy [1]. □

## References

[1] G. Nemhauser, L. Wolsey, and M. Fisher. An analysis of approximations for maximizing submodular set functionsi. *Mathematical Programming*, 14(1):265–294, 1978.