[Reviews · NeurIPS 2015]

Submitted by Assigned_Reviewer_1

I don't know that I find the evaluation set up all that convincing. From what I gather, there sensor costs are uniform.

Thus the authors show a plot of features vs. error which makes it more like a feature selection problem than budgetted learning. Non uniform costs seem much more realistic and would conform more to the motivation given in the intro. The goal isn't to minimize the number of features, but the total cost of the features. (Note there are some typos in the sentence about costs and this should could be more precise).

Is the proposed approach siginficantly better than the two baselines on the large sensor set?

Minor points *Paper is quite dense *The paper also ends a bit abruptly *ERM isn't defined before it is used

#####After rebuttal#####

I understand it is non trivial to come up with cost structures and assuming uniform cost is a standard. Still, at some point it would be good to go beyond this standard assumption. Surely many domains must have non-uniform costs, e.g., medical tests (e.g., MRI > x-ray). Even doing so on a synthetic set up at first would be a nice contribution.
Summary: The paper has a nice mix of theory and practice, though I have some questions about the experiments.

Submitted by Assigned_Reviewer_2

This paper proposes a novel adaptive sensor acquisition system learned using labeled training examples, where

the system is modeled as directed acyclic graph (DAG). DAG structures are learned by using dynamic programing for minimizing an empirical risk. In addition, the method has a theoretical guarantee that the expected risk converges to the Bayes risk when the number of examples goes to infinity.

Experiments using bench mark datasets shows the proposed method has a super performance with respect to accuracy.

I'm not an expert of this problem, but the proposed method is interesting.

A very well- written paper, a pleasure to read. No obvious flaws.

Minor comments

Line 37; atypical cases -> a typical case Line 110; To model, the acquisition process -> To model the acquisition process Line 171, pass thru -> pass though
Summary: the proposed method is interesting. A very well- written paper, a pleasure to read. No obvious flaws.

Submitted by Assigned_Reviewer_3

This paper proposes an algorithm for reducing test-time feature acquisition cost. Instead of jointly optimizing a complex cascade/tree structured classifier, the authors introduce an approach that efficiently learns the structured classifier through dynamic programming and reinforcement learning. The paper also gives a convergence analysis of the proposed algorithm.

In general, this paper is well written and easy to follow. I enjoy reading the idea of using policy learning and dynamic programming to efficiently learn the classifier. Moreover, compared to most of the previous work, which is less desirable because of the convergence only to fixed point, this work shows that the algorithm converges to the Bayes risk asymptotically. The concern is in the result section. Given the proposed idea is similar to the R. Busa-Fekete et. al., which also uses DAG and reinforcement learning, the authors should compare against this work. Without a comparison with this work is my main reason for lowering my score.
Summary: An good paper with clear advancement in budgeted learning. However, the result section is incomplete.

Submitted by Assigned_Reviewer_4

Resource constraint prediction has to select the smallest/ cheapest set of sensors in order to make accurate decisions. A new approach to this problem is proposed in this paper where the authors basically define a deterministic Markov decision process and learn its policy applying cost sensitive learning in every state of the MDP. Given a new testing example to classify, the policy has to decide

whether a new sensor should be queried (if yes, which sensor), or whether the prediction should be made with the current set of sensors measured.

In line 44, the authors mention the "expected budget constraint". Please be clear about that because your paper is not budgeted learning; budgeted learning is more challenging, please see, e.g. papers of Russell Greiner (University of Alberta) on this topic.

The DAG used in this paper defines an MDP. Perhaps a closer connection with MDPs could be made, e.g., you could mention that this is a special MDP that can be solved efficiently using your method. Also, it would be good to explain what are the key techniques/assumptions that allow solving this MDP efficiently. Putting your method in a broader MDP context would be very useful.

Line 122, a quotation mark is missing.

I have some problems understanding equation (5). If (s_j,s_k) is and edge, how can I read from the first case in the equation (5) that this represents the cost of acquiring a new sensor? Please explain. Shouldn't there be $t\in s_k \ s_j$ ?

One of my key concerns is R that is used in Lemma 2.1. In [2], the authors use importance-weighted binary classification. This is basically classification that considers misclassification cost. What I am missing in Lemma 2.1 and its proof is a clear link between misclassification cost in [2] and the cost of features (or sets of features) in Lemma 2.1. Please explain. I have another problem with lemma 2.1. If node j leads to a leaf then the cost of sensors is zero because there is no need to pay for any sensors; why then, the cost of an edge j->k is considered in the proof of lemma 2.1.

Vectors w are not defined in eq. 8. Well, (2) in Alg. 1 says "construct the weight vector w of edge costs per action". Please explain how to do that.

It looks that the key element in the analysis in section 2.3 is to optimise at each stage over all examples in the training set. It sounded weird to me initially, but I believe that that's correct.

My second main concern are Lemma 3.1 and theorem 3.2. As long as sub-modularity in Lemma 3.1 is correct I believe, I am not sure why we should not consider sub-modularity in the theorem 3.2 as well. The risk in the theorem is monotonically decreasing indeed, but we should really talk about diminishing returns which don't exist because adding a new feature f_1 may not change anything until we add another features f_2 when those two features interact. So, there are not diminishing returns and sub-modularity does not apply. Additionally, in the proof of theorem 3.2 the authors consider adding a new subset (line 34 in the appendix), but doing that will eventually violate the constraint in e.q. (1) in the appendix. I believe that this is another reason why this problem is not sub-modular, and the greedy approach can be very bad.

Also, "the reward" in lemma 3.1 should

be explained.
Summary: A the first glance this seems to be a solid paper with a strong contribution, but a few things could be explained in a better way. Also, it is not clear if all the claims are correct. Hopefully, the authors' response will provide missing explanations.

Author Feedback
Author rebuttal: We thank the reviewers for the comments and will incorporate the suggestions.

***Reviewer 2: Lower Rating because of lack of comparison to Busa-Fekete 2012

*Citation of Busa-Fekete:
Note that we do cite Busa-Fekete's work ([4]). While it is related to the general literature of learning & budgets we do not see how to extend [4] to our problem of test-time feature acquisition with feature acquisition costs.

*Issues with comparing to Busa-Fekete Algorithm:
Recall the goal of [4] is to reduce the number of weak learner evaluations during test-time. The weak learners are obtained from a boosting algorithm. Their ordering is a priori fixed with the order determined by their index. As such there is no notion of sensor acquisition cost. The DAG structure in Busa-Fekete et al. arises from a skip options added to this fixed-order cascade, with each node corresponding to a weak learner.

It does not appear to be straightforward to extend [4] to include sensor costs and sensor order selection. This is particularly because different weak learners could use the same sensor and the costs for using these weak learners would have to be discounted.

Furthermore, in contrast to [4], we propose a scheme to learn decisions in general DAG structures, with each node corresponding to a subset of sensors. In this case, no known order exists and sensor acquisition costs must be accounted for. Algorithmically, the approach of Busa-Fekete et al. 2012 models the reward of each action to solve the MDP, whereas our proposed approach learns decisions directly.

*CSTC as a Generalization of Busa-Fekete:
In this context, it is worth noting that the CSTC algorithm proposed by Xu et al. can be thought of as a generalization of Busa-Fekete with explicit sensor costs and more flexible structure. Note that we compare it to our method in the experimental results (see Sec. 4.2). Indeed, an important aspect of Xu et. al.s work is in accurately discounting feature costs for a weak learner proposal based on previously acquired features in the context of an already utilized weak learner.

***Reviewer 3:

**Equation 5 typo:
We apologize for the typo in Eqn 5, the correct argument of the summation is $t \in s_k \ s_j$, that is sensors in subset s_k that are not in subset s_j.

**Connection between policy error and feature costs:
In Lemma 2.1, the classification decisions in the cost-sensitive learner (as learned in [2]) translate to actions selected by the policy $\pi_j$. Misclassifications in the cost-sensitive learner result in non-optimal actions. The cost of the action selected by the policy is defined in the proof of Lemma 2.1, and includes classification error and/or sensor acquisition cost.

**Costs of edges leading to leafs:
The risk R used in Lemma 2.1 incorporates the cost of misclassification if the edge leads to a leaf node (note that the only leaf node in the graph is s_{SC}). From the second case in Eqn 5, if the edge leads to the leaf node, the edge cost is the classification error of the example using the sensor subset s_j. As the algorithm progresses, the edge costs of the reduced graph are updated to include the costs below the edge in the graph, and in this manner, the cost of misclassification is combined with sensor cost.

**Definition of weight vectors:
The weight vector w_i is simply a concatenation of the outgoing edge costs for example x_i, with the kth element of w_i equal to C(x_i,y_i,s_j,s_k).

**Supermodularity typo:
We apologize for the typo on the statement of Theorem 3.2 and thanks for pointing it out. We will correct it in the revised version. The classification function $f$ is assumed to be supermodular and monotonically decreasing, and therefore the reward is submodular and monotonically increasing. This allows us to show the constant order approximation for the greedily constructed subsets.

**Violation of subset constraint:
In the proof of Theorem 3.2, we point out that the reward induced by adding a sensor to a subset is equivalent to the reward for having both subsets in the collection of subsets. This is because the reward is monotonically increasing (that is, we assume that adding a sensor does not decrease performance). As such, adding a sensor to a subset can be viewed as adding that subset to the collection of subsets, though the actual collection of subsets does not increase in size. We will explain this in the revised paper.

***Reviewer 5:
**Uniform sensor costs
We use uniform feature costs in the experimental section following the precedent set in literature, notably Wang et al., Xu et al., and Kusner et al., against whom we compare performance. The issue is that there are very few datasets for which one can propose natural cost-structures. Hence uniform costs have come to be accepted as surrogates.

**Abrupt Ending:
Thanks for pointing it out. We will include a conclusion and future work section in the revised version.